ecology, health and disease and epidemiology, evolution

epidemic fadeout, lyssavirus, Neotropics, reservoir, spatial structure, zoonosis

**Authors for correspondence:**
Daniel G. Streicker
e-mail: daniel.streicker@glasgow.ac.uk
Bernal Leon
e-mail: bleon@senasa.go.cr

# Phylodynamics reveals extinction–recolonization dynamics underpin apparently endemic vampire bat rabies in Costa Rica

Daniel G. Streicker[1,2], Silvia Lucia Fallas González[3], Giovanna Luconi[4], Rocío González Barrientos[5,6] and Bernal Leon[5]

[1]MRC—University of Glasgow Centre for Virus Research, Glasgow, UK
[2]Institute of Biodiversity, Animal Health and Comparative Medicine, College of Medical, Veterinary and Life Sciences, University of Glasgow, Glasgow, UK
[3]Laboratorio de Pruebas de Paternidad, Caja Costarricense del Seguro Social, San José, Costa Rica
[4]Clínica Veterinaria Clivet, San José, Costa Rica
[5]Laboratorio de Bioseguridad, Laboratorio Nacional de Servicios Veterinarios, Servicio Nacional de Salud Animal, Heredia, Costa Rica
[6]Department of Biomedical Sciences, College of Veterinary Medicine, Cornell University, Ithaca, NY, USA

DGS, 0000-0001-7475-2705

Variation in disease incidence in wildlife is often assumed to reflect environmental or demographic changes acting on an endemic pathogen. However, apparent endemicity might instead arise from spatial processes that are challenging to identify from traditional data sources including time series and field studies. Here, we analysed longitudinal sequence data collected from rabies virus outbreaks over 14 years in Costa Rica, a Central American country that has recorded continuous vampire bat-transmitted rabies outbreaks in humans and livestock since 1985. We identified five phylogenetically distinct lineages which shared most recent common ancestors with viruses from North and South America. Bayesian phylogeographic reconstructions supported bidirectional viral dispersals involving countries to the north and south of Costa Rica at different time points. Within Costa Rica, viruses showed little contemporaneous spatial overlap and no lineage was detected across all years of surveillance. Statistical models suggested that lineage disappearances were more likely to be explained by viral extinctions than undetected viral circulation. Our results highlight the importance of international viral dispersal for shaping the burden of rabies in Costa Rica, suggest a Central American corridor of rabies virus invasions between continents, and show that apparent disease endemicity may arise through recurrent pathogen extinctions and reinvasions which can be readily detected in relatively small datasets by joining phylodynamic and modelling approaches.

## 1. Introduction

Many newly emerging and historically important human pathogens originate from wildlife [1]. Understanding how these pathogens circulate in their natural hosts and what drives cross-species emergence remains a major challenge for epidemiology and public health [2]. Existing datasets typically comprise short-term field studies and longer-term, but sparser, records of cross-species transmission to humans or domestic animals detected by surveillance systems [3]. Although these data can reveal epidemiological patterns, markedly different underlying ecological processes can be responsible. Variation in incidence can arise from extrinsic environmental drivers (e.g. variation in climate or food availability that influences host immunity or population dynamics) or intrinsic

demographic factors (i.e. seasonal replenishment of susceptible individuals) [4,5]. Alternatively, apparent cycles of endemic infection may reflect recurring extinction–recolonization dynamics of the pathogen. The best examples of this phenomenon come from highly monitored human and domestic animal pathogens (e.g. dengue virus and foot and mouth disease virus), where cycles of extinction and replacement of antigenic types influence vaccination strategies [6,7]. Many wildlife pathogens also persist through spatio-temporal processes, but except in rare cases (e.g. invasions of highly visible diseases such as white nose syndrome in bats or chytridiomycosis fungus in amphibians), the dynamics of spatial spread are unobservable due to the scarcity of genotyped or serotyped longitudinal incidence data [8,9]. Unrecognized extinction–recolonization dynamics potentially mislead conclusions on the mechanisms of pathogen persistence. Favoured models would incorrectly predict long-term, localized persistence in the absence of immigration, leading to biased estimate the basic reproductive number ($R_0$), underestimation of the critical community size required for pathogen persistence or failure to appreciate immune-mediated competition among strains [10,11]. Consequently, interventions or risk forecasts based on these models will less effective.

Vampire bat (*Desmodus rotundus*) rabies virus (VBRV) exemplifies a zoonosis for which limited understanding of transmission dynamics in a wildlife reservoir poses substantial management challenges. Although VBRV was discovered over a century ago, it remains the primary cause of uniformly lethal human and animal rabies outbreaks throughout most of the Latin American tropics and subtropics [12]. Losses to livestock production exceed US$50 million annually across Latin America, not including major investments in prevention, surveillance and control [13]. Human VBRV infections appear to be rising such that vampire bats routinely feed on humans [12]. Efforts to limit the burden of VBRV include vaccination of humans and livestock; however, the unpredictable nature of VBRV outbreaks, together with high costs of vaccines, limit widespread preventive use [14,15]. Vampire bats have also been culled using anticoagulant poisons since the 1970s, but practically attainable levels of population reduction appear to have negligible effects on rabies risk, in part due to the weak relationship between bat colony size and rabies incidence [16]. Better understanding of VBRV spatio-temporal dynamics could therefore open new opportunities for anticipatory vaccination of humans and domestic animals or epidemiologically informed control within the bat reservoir.

Most knowledge of the transmission dynamics of VBRV is derived from large countries in North (i.e. Mexico) and South America (e.g. Argentina, Brazil, Peru) that sustain high incidence of human and/or livestock rabies [17–19]. These studies have revealed that single bat colonies cannot maintain VBRV indefinitely, implying that bat dispersal among colonies enables long-term viral persistence [20,21]. However, distinct underlying spatial processes, including travelling waves of infection into historically rabies-free areas and metapopulation dynamics in endemically infected areas, have been identified [22–24]. A key remaining knowledge gap for VBRV, and rabies virus more generally, is the spatial scale required to evade stochastic viral extinction. The Central American tropics comprise a relatively narrow land mass, which might predispose spatially dependent pathogens to extinction. If so, apparent endemic circulation in some countries might be driven by re-introductions from North and/or South America.

Alternatively, if the geographical scale required for viral persistence is sufficiently small or if environmental conditions favour sustained viral circulation, VBRV might persist endemically. These scenarios imply different strategies and opportunities for prevention and control.

Little is known of the ecology of VBRV circulation in Central America, although rising human rabies incidence has been noted [12,25,26]. Here, we present the first molecular epidemiological study of VBRV in Central America, focusing on Costa Rica, where the virus was first reported in 1968, and rabies outbreaks in humans and livestock have been documented annually since 1985 [27]. Specifically, using virus sequence data obtained from domestic animals that succumbed to rabies over a 14-year period, we (i) identified the time scale of evolutionary relationships between VBRVs from Costa Rica and other North, Central and South American countries, (ii) reconstructed the histories of international and inter-continental viral dispersals through Central America, and (iii) examined whether the dynamics of viral lineage residence in Costa Rica were better explained by endemic circulation of resident viruses or extinction–recolonization dynamics.

## 2. Methods

### (a) Rabies virus data and sequencing

Over the time period of this study (2004–2017), 60 rabies outbreaks were reported in domestic animals from Costa Rica, involving 180 mortalities (median = 2 deaths per outbreak; range = 1–17; electronic supplementary material, figure S1). All animals were reported by their owners to have clinical signs consistent with rabies encephalitis, and brain samples were confirmed rabies positive by direct immunofluorescence by the National Veterinary Service Laboratory (LANASEVE). We sampled representative viruses collected from 39 cattle and 1 domestic dog from 38 outbreaks. Importantly, as cattle do not transmit rabies intra-specifically or back to bats, each record represents an independent spillover from the bat reservoir. The single rabies virus from a domestic dog (GenBank accession no. KU550098) was also from a vampire bat-associated lineage [27]. The number of viruses sampled per year was correlated with the number of outbreaks reported, indicating the stability of the sequencing effort through time ($r = 0.70$, $p = 0.006$).

Upon arrival to LANASEVE, samples were stored at −80°C. RNA extractions used the DNeasy Blood and Tissue Kit (Qiagen), following the manufacturer's instructions. Reverse transcription–polymerase chain reaction (RT–PCR) used the primers RAB20 5′ ACGCTTAACAACAARATCARAG-3′ and RAB304 5′-TTGACGAAGATCTTGCTCAT-3′ targeting the complete nucleoprotein gene [28]. The nucleoprotein is an informative gene for phylogeographic analyses of rabies and is the most widely sequenced VBRV gene in GenBank (2440 records versus 596 in the glycoprotein and fewer in other genes; accessed 31 July 2019 via http://rabv.glue.cvr.ac.uk), which maximized our ability to detect incursions into or out of Costa Rica [18,29]. One-step RT–PCR was performed with the following conditions: 45°C for 30 min, 95°C for 15 min, then 40 cycles of 94°C for 10 s, 53°C for 45 s, 68°C for 1:30 min, followed by 68°C for 10 min. The 1534 base pair amplicon was visualized under UV light after electrophoresis on 1% agarose gels containing GelRed Nucleic Acid Stain (Biotium) and a slice of the gel was purified with QIAquick Gel Extraction Kit (Qiagen). The sequencing reaction was performed with BigDye TERMINATOR v. 3.1 using the following cycle conditions: 30 cycles of 96°C for 10 s, 50°C for 5 s and 60°C for 4 min; products were purified with BigDye XTERMINATOR. DNA sequencing was performed on a 3130 Genetic Analyzer (Applied

Biosystems). Sequences were aligned using MAFFT and trimmed to the coding regions [30].

## (b) Viral phylogenetic analysis

Analyses used two datasets comprising sequences from Costa Rica with (hereafter 'international dataset', $n = 75$) and without ('national dataset', $n = 40$) sequences from other North, Central and South American countries. International analyses focused on the time scale and geographical patterns of viral dispersals into and out of Costa Rica using discrete phylogeographic analyses. We used a cut-off of 98% similarity to any Costa Rican sequence to identify VBRVs available in GenBank that could plausibly have shared a most recent common ancestor (MRCA) with Costa Rican viruses, but also included additional representative VBRV lineages for reference. National data were used to confirm the geographical origins suggested in the international analyses and to examine patterns of viral diffusion within Costa Rica using continuous phylogeographic analyses.

Both datasets had evidence of clock-like evolution according to TempEst (international: slope = $5.56 \times 10^{-4}$, $r = 0.75$; national: slope = $7.08 \times 10^{-4}$, $r = 0.70$). Preliminary phylogenetic analyses of the international dataset in BEAST v. 1.8.4 used stepping stone sampling to estimate the marginal likelihood and Bayes factor (BF) support for the strict, relaxed lognormal and random local clock models (electronic supplementary material, file S1) [31,32]. The relaxed lognormal molecular clock was indicated (BF > 7.5 and 35.8) and used in subsequent analyses using a normally distributed prior (mean = $5.6 \times 10^{-4}$, s.d. = $2 \times 10^{-4}$). Molecular clock models were not formally compared for the national dataset since the random local clock cannot currently be applied to single analyses with multiple trees (see below). We used a relaxed lognormal clock (uniform prior from $1 \times 10^{-4}$ to $7 \times 10^{-4}$) since substantial rate variation among branches (ucld.stdev > 0.65) rejected the strict molecular clock. We used partitionFinder2 to identify the best fitting models of codon partitioning and nucleotide substitution, according to the Akaike information criterion corrected for small sample size [33]. International analyses used a TRN + I + G model shared by codon positions 1 and 2 and the GTR + G model for codon position 3. National analyses used the HKY model shared by codon positions 1 and 2 and a TVM model with equal base frequencies for codon position 3. Customized nucleotide substitution models were implemented in BEAST by manually editing the xml file generated by BEAUTI.

Discrete phylogeographic modelling in BEAST used country of viral origin as a categorial trait and was set to infer asymmetric rates between countries [29]. Models were run for 150 million generations with trees sampled every 10 000 generations (electronic supplementary material, file S2). Given the potentially complex demographic histories within the international dataset, we specified the Bayesian skyline demographic model as a flexible prior of viral effective population size through time [34]. Bayesian stochastic search variable selection was used to estimate BF support for transitions between countries, and discrete Markov jumps recorded the magnitude of strongly supported transitions. Continuous phylogeographic models used the latitude and longitude of samples to reconstruct historical diffusion within Costa Rica [35]. Costa Rican viruses had different geographical origins which would have obscured reconstructed ancestral locations, but the limited number of sequences precluded independent analyses of each lineage. We therefore modelled each of the two largest monophyletic Costa Rican lineages ($n = 13$ and 14 sequences, respectively) as separate data partitions in the same BEAST analysis [36]. Specifically, tree topologies and spatial reconstructions were independent, but evolutionary models and parameters were shared across lineages (electronic supplementary material, file S3). Given the limited data, this analysis used simpler demographic models of constant viral effective population size. Among four models of spatio-temporal diffusion, the gamma-distributed rates model was favoured over Brownian motion, Cauchy distributed and lognormal distributed rates (BF > 8.6–15.1; electronic supplementary material, file S1). Dispersal velocities (mean branch velocity) were calculated using 1000 randomly sampled, post burn-in trees from the posterior using the *Seraphim* package in R [37,38]. We modelled 1 year of uncertainty on the date of samples for which only the year of collection was available (precision = 1). We used TRACER v. 1.7.1 to verify parameter convergence, check that effective sample sizes on parameters exceeded 200 and select appropriate burn-ins.

## (c) Temporal analysis of putative national viral invasions and extinctions

We evaluated evidence for viral invasions and extinctions given the possibility of undetected viral circulation using two distinct approaches. First, we tested whether the probability of observing each viral lineage varied through time by fitting a multinomial logistic regression using the *multinom* function in the *nnet* package of R. Specifically, we modelled the identity of the viral lineage associated with each rabies outbreak as a nominal dependent variable with five levels (the number of discrete virus lineages detected), with year as the only covariate. This analysis used the earliest detected virus (L2) as a baseline outcome against which to compare other lineages ($i$), leading to

$$\ln\left(\frac{P(\text{virus} = i)}{P(\text{virus} = \text{L2})}\right) = b_{0,i} + b_{i,\text{L2}}\text{year},$$

where $b_{0,i}$ is the intercept for virus $i$ and $b_{i,\text{L2}}$ is log odds effect of the change in year on an outbreak being caused by virus $i$ versus virus L2. Results were qualitatively similar for all alternative baselines. We calculated a pseudo-$r^2$ and $p$-value for the effect of year by comparing the log likelihood of the fitted model against an intercept-only model using the *pscl* package. Second, we used the number of months between successive detections of the same viral lineage (interval data) to assess evidence for continued viral circulation at subdetectable levels versus extinction. We compared the intervals between successive detections of each virus while it was known to have circulated (based on genetically typed outbreaks) to the interval between the final detection of each virus and the end of the observation period, capped by the last outbreak in our dataset (2017.66). Values for the time since final detection are therefore conservative lower bounds, representing minimum estimates. The empirical probability density of inter-detection intervals was estimated using the *demp* function in the *EnvStats* package of R, after aggregating intervals observed across all viruses. To avoid artificially reducing the distribution of inter-detection intervals, we removed repeat observations of the same virus in the same month (i.e. intervals of zero). The *pemp* function was used to estimate the probability of virus lineage extinction, given the number of months since its final detection. R scripts for both statistical analyses are provided in electronic supplementary material, file S3.

## 3. Results

Phylogenetic analysis of the international dataset revealed two main paraphyletic VBRV lineages that circulated in Costa Rica between 2004 and 2017 (figure 1*a*). Across the entire tree, the VBRV nucleoprotein evolved at a median rate of $4.86 \times 10^{-4}$ substitutions per site per year (95% highest posterior density (HPD) = $3.54$–$6.34 \times 10^{-4}$), which was similar to previous estimates [18]. Lineage 1 (L1) was encountered between 2006 and 2017 and had a MRCA in 1980.8

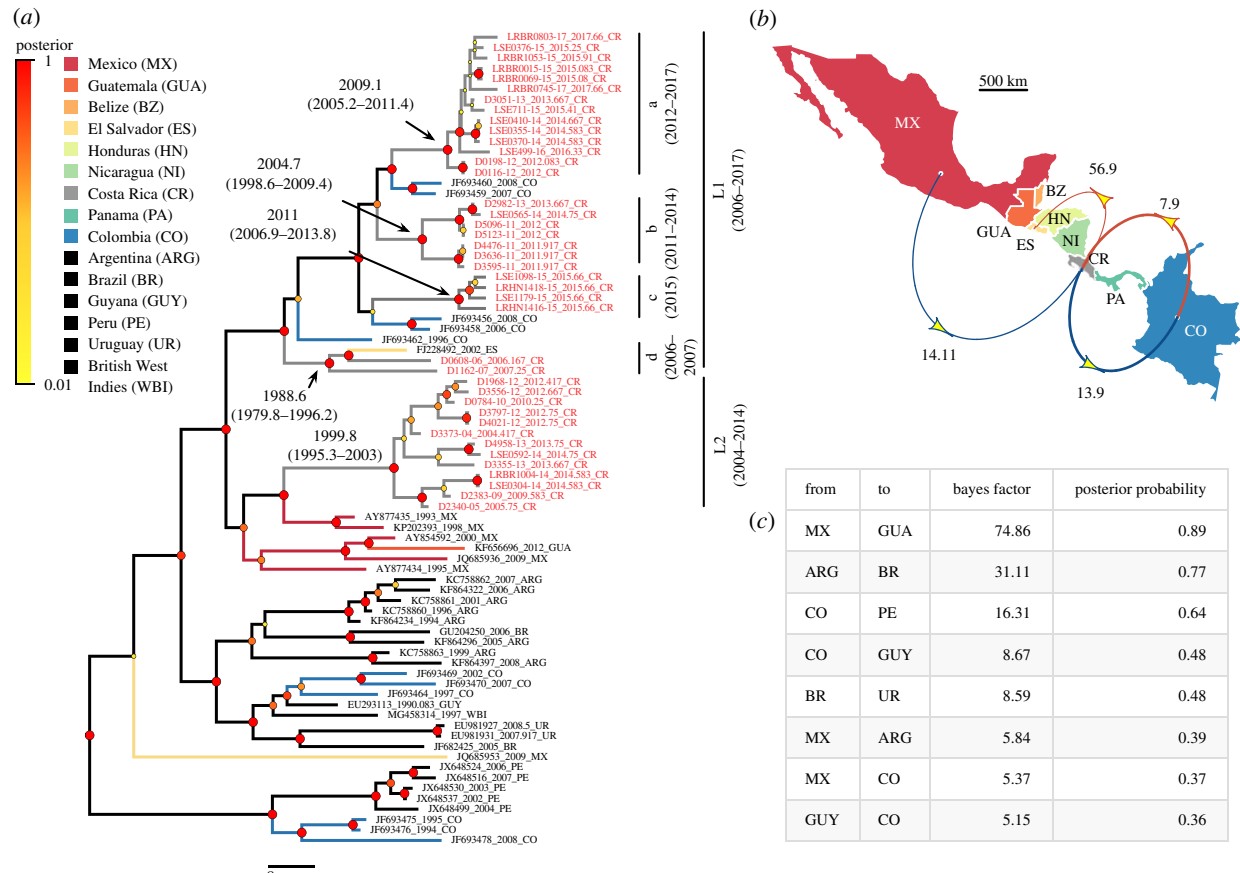

**Figure 1.** Bayesian phylogenetic inference of international dispersal histories of vampire bat rabies. (*a*) Phylogenetic tree with branches coloured by the country of origin. Black branches signal countries omitted from the map in (*b*). Node circle size and colours indicate posterior probabilities on branch partitions. Red tip labels indicate viruses from Costa Rica. (*b*) Summary of statistically supported (BF > 5) viral dispersions between countries, restricted to those involving Costa Rica. Arrow colours represent northbound (red) or southbound (blue) dispersals, with width proportional to the number of Markov jumps (range = 0.9–2.2). Arrows are annotated with the BF support for dispersals between countries. (*c*) Additional transitions supported in the discrete phylogeographic analysis that did not involve Costa Rica occurred predominantly between geographical neighbours. Country name abbreviations in (*b*) and (*c*) match (*a*). (Online version in colour.)

(95% HPD = 1970.5–1989.33). However, L1 comprised four temporally structured clades, several of which shared an MRCA with South American viruses from Colombia, rather than the other Costa Rican viruses (figure 1*a*). Consequently, the discrete phylogeographic analysis identified repeated bidirectional viral dispersal between Costa Rica and Colombia (figure 1*b*). This analysis further supported a dispersal of an L1 virus from Costa Rica to El Salvador, consistent with the spread of South American viruses to more northern Central American countries via Costa Rica. Lineage 2 (L2) was detected between 2004 and 2014 and shared an MRCA with Mexican viruses in 1999.8 (95% HPD = 1995.3–2003). The phylogeographic analysis supported unidirectional viral dispersal from Mexico to Costa Rica, suggesting the southward invasion of a North American virus (figure 1*b*).

Ancestral state reconstructions of the latitude and longitude of the progenitors of the L1a and L2 viruses (the two Costa Rican viruses with sufficient sample sizes for this analysis) provided additional support for invasions suggested by the discrete phylogeographic analysis. The inferred ancestor of virus L1a occurred in southern Costa Rica near the border with Panama, consistent with a recent invasion from South America (figure 2). This invasion spread northward at an average of 11.61 km yr$^{-1}$ (95% credible region on the mean branch

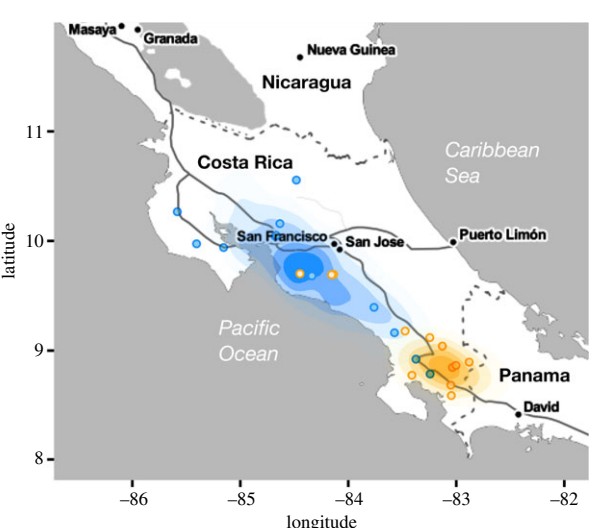

**Figure 2.** Reconstructed geographical origins of two vampire bat rabies viruses encountered in Costa Rica. Map shows kernel density plots of the inferred ancestral latitude and longitude of the L1a (orange) and the L2 (blue) viruses, each estimated from 1000 randomly selected trees from the posterior distribution of the continuous phylogeographic analysis. Points show the observed locations of outbreaks attributed to each virus. (Online version in colour.)

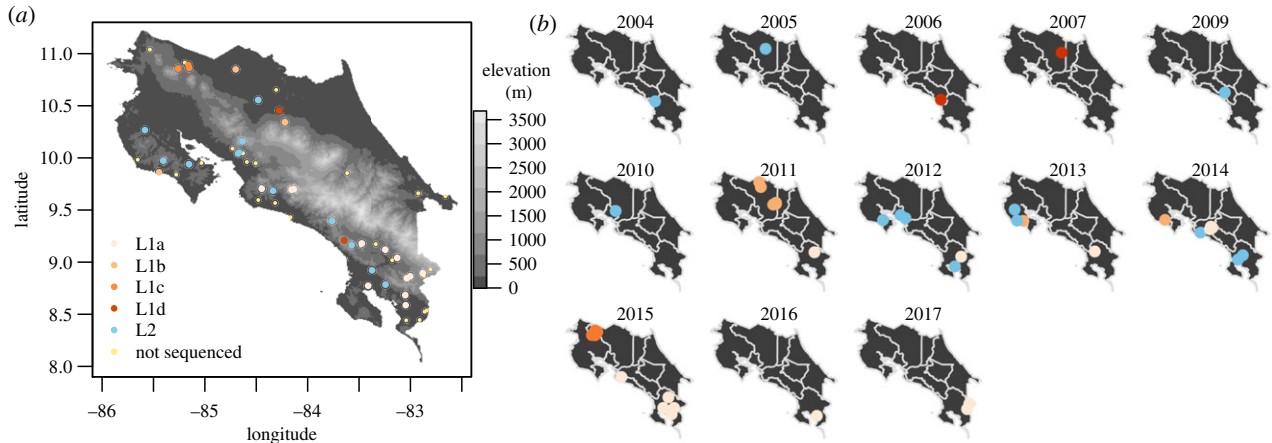

**Figure 3.** Spatio-temporal dynamics of vampire bat rabies within Costa Rica. (*a*) The location of all samples coloured by phylogenetic group. Non-sequenced outbreaks between 2004 and 2017 are shown in light yellow (see electronic supplementary material, figure S1 for the spatio-temporal distribution of all outbreaks from 1985 to 2018). (*b*) Snapshots showing the viral presence through time (colours as in (*a*)). The year 2008 was omitted because no sequence data were available from a single outbreak during that year. Points are jittered to minimize overlap. (Online version in colour.)

velocity = 4.08–59.93) while in Costa Rica. The inferred ancestor of virus L2 occurred further north in central Costa Rica, but was considerably less precise, suggesting that the longer time period of circulation in Costa Rica largely erased evolutionary signal of viral origins (figure 2). This led to higher and more variable velocities which should be interpreted cautiously (median = 56.98 km yr$^{-1}$, 95% credible interval = 12.62–2522.36).

Within Costa Rica, viruses showed variable degrees of temporal residence and spatial structure, and no virus lineage was detected across the duration of the study (figure 3). For example, L1c appeared in 2015 near the border with Nicaragua and was not subsequently observed, and L2, which was geographically widespread from 2004 to 2014, was not detected in 2015–2017 (figure 3*b*). Indeed, during the final 2 years of observation (2016–2017), only L1a viruses were encountered and cases occurred exclusively in the south of the country. Despite the relatively small sample sizes underlying the lineage-specific time series, the multinomial regression found an effect of year on which virus was observed ($\chi^2 = 29.6$, $p < 0.001$, McFadden's pseudo-$r^2 = 0.26$) and had sufficient power to detect viral invasions and extinctions, evidenced by 95% CIs on the predicted probability of viral presence through time rising above or approaching zero, respectively (figure 4*a*). L1a and L1c were unlikely to have circulated in Costa Rica during the first 6 years of the study (mean probability of presence less than 0.01) and viruses L1d, L1b and L2 had a high probability (0.975–0.999) of having gone extinct by 2017 (figure 4*a*). Rabies outbreaks from 2018 (the year after our sequencing completed) revealed no additional cases in other regions, suggesting the continued absence of other viral lineages (electronic supplementary material, figure S1). By the end of the molecular study, non-L1a viruses had not been encountered for 2.96 years on average (range: 2–10.41 years), which was longer than the expected interval between successive detections of persisting viruses (figure 4*b*). The estimated probabilities of failing to detect a still-circulating virus were less than 0.005 for viruses L1b, L1d and L2, and 0.11 for virus L1c, confirming that all viruses other than L1a were likely to have gone extinct from Costa Rica. By combining molecular clock estimates of the time since the MRCA of clades with the final dates of viral detections, we estimated viruses circulated in Costa Rica for only 10.5 years on average (range: 4.7–18.65 years).

## 4. Discussion

Distinguishing endemic pathogen circulation from extinction–recolonization dynamics is a major challenge which has potential to undermine strategies for disease prevention and control. Focusing on Costa Rica, a small Central American country that has detected rabies nearly every year for decades, we observed invasions of North and South American viruses as well as northbound and southbound exportations of VBRV; however, all viruses had relatively brief residence times within the country. These observations are most parsimoniously explained by a dynamic process of spatial invasion fronts among Central American countries, each of which has limited capacity to sustain enzootic transmission in isolation. Consequently, patterns of spillover to humans and livestock reflect epidemiological processes occurring across multiple countries, which creates challenges and opportunities for prevention and control.

Apparent viral extinctions and reinvasions might alternatively arise from cryptic, enzootic circulation of viral lineages, with periodic detection driven by fluctuating incidence in bat populations or inconsistent surveillance effort through time. Indeed, a variety of factors have been suggested to alter VBRV incidence in bats or spillover to humans or livestock, including temperature and precipitation associated with El Niño oscillations, changes in prey availability or quality, and age-structured transmission [21,39–41]. Here, phylogeographic signals of international viral spread, temporal structuring of viral clades and statistical evidence for prolonged absences of each virus lineage at different time points make cryptic viral persistence unlikely. Moreover, VBRV disappeared from the entire Caribbean coast of Costa Rica in 2008, suggesting an extinction over a large geographical scale which lasted at least 10 years (electronic supplementary material, figure S1). Given the characteristic clinical signs of rabies in cattle (hind limb paralysis, rapidly followed by death), it is unlikely that outbreaks would be completely unnoticed over a decade if they had occurred. An important consequence of extinction–reinvasion dynamics is that environmental or demographic factors alone cannot directly trigger spillover by increasing bat population density, reducing immunity or altering feeding behaviour since the virus will often not be locally present to respond to these potential drivers. Increases in transmission

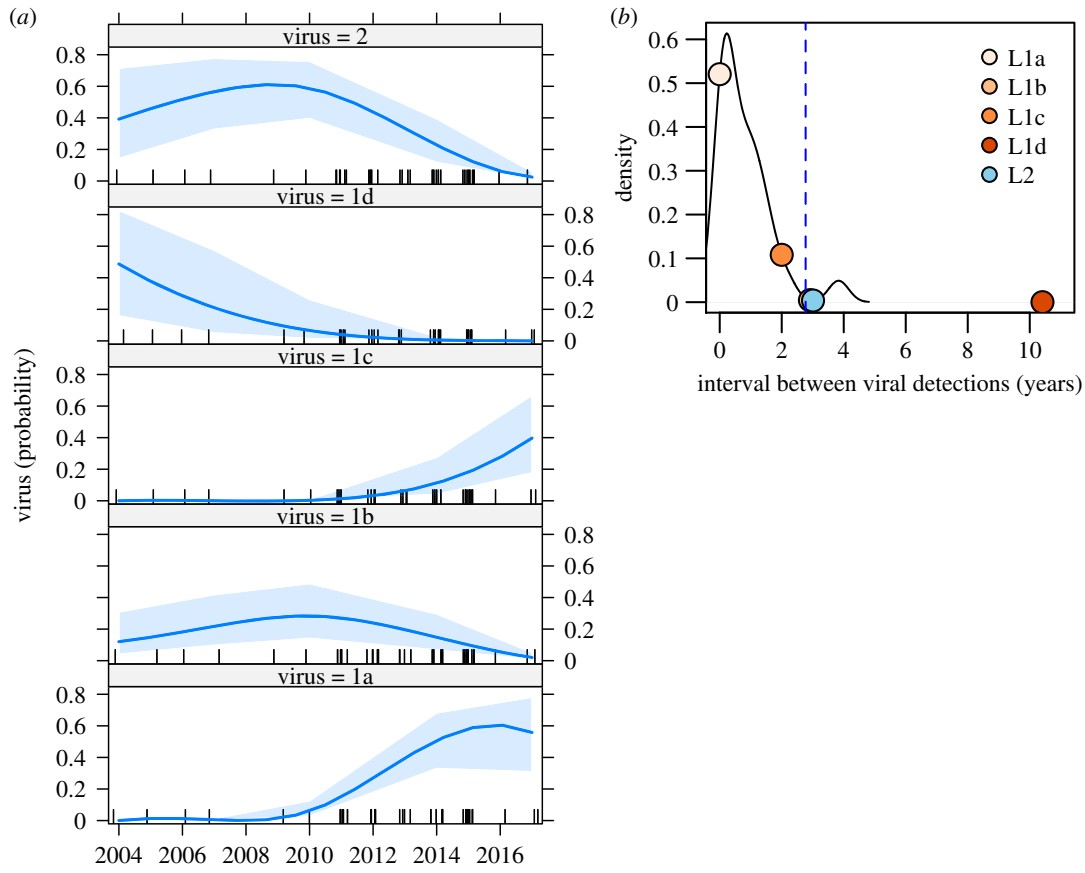

**Figure 4.** Invasions and extinctions of five viral lineages. (*a*) The predicted probability of viral presence from 2004 to 2017 for each viral lineage. Dashes across the *x*-axis indicate observations of any virus; the shading indicates the 95% CI on the mean probability (solid blue line). (*b*) The probability density of intervals between successive observations of the same viral lineage. The blue dashed line is the 95% upper bound. Points indicate the duration of the interval between the final detection of each virus (L1b, L1c, L1d, L2) and the final VBRV detection in the study (virus L1a in 2017.66). (Online version in colour.)

might instead be explained by environmentally or anthropogenically driven increases in bat dispersal that connect VBRV-free to nearby VBRV-infected populations [20]. For example, seasonal births create pulsed availability of susceptible vampire bats, but few studies have detected seasonal spillover to livestock [24,39,42,43]. Our findings reveal that birth pulses will only be important in times and places when the virus is present or if they promote mixing of infected and uninfected colonies. Consistent with the latter possibility, seasonal spatial expansions of VBRV have been attributed to annual expulsions of maturing male bats from maternity roosts [18].

Locally sporadic viral presence also helps explain why bat culls have not reduced VBRV spillover [21,44]. Since culls are carried out haphazardly or in response to increasing frequencies of bat attacks on humans or livestock, they are unlikely to affect areas of active viral circulation or high-risk areas for VBRV invasion by chance. Therefore, culls mostly reduce VBRV-free populations rather than preventing endemically infected populations from reaching a critical epidemic threshold as would be required for culling to succeed [10]. Designing culls (or emerging alternatives including oral vaccines) to limit invasions and potentially trigger lineage extinctions might be more effective and less expensive than the current approach [45]. Given the importance of international viral dispersal demonstrated here, efforts in Central America might target narrow geographical pinch points, analogous to the combined use of natural landscape and vaccine barriers to block the spread of raccoon rabies in North America [46]. As less than 3% of vampire bats disperse

and median dispersal distances are less than 10 km, such interventions might be carried out over relatively small widths [42]. Other migratory bat species might spread rabies over larger distances, but maintain epidemiologically independent viruses that would not compromise interventions targeting VBRV [19]. Increasing notifications to surveillance systems and data sharing across Latin America is a key prerequisite to design epidemiologically informed interventions.

Our findings also illustrate a Central American corridor of VBRV between North and South America. Inter-continental spread of VBRV was previously implied by large-scale phylogenetic evidence that viral variants occur on both continents, but whether this reflected ancient or modern spread was unknown [19]. The recent time scale of the dispersal events we observed (within the last 30 years) argues strongly that this corridor is sustained in the present day. Although we did not observe a full sweep across continents (i.e. a North American virus traversing Costa Rica to arrive in South America or vice versa), this was expected, given that the sparse sampling of viruses and the relatively slow progression of rabies expansions in vampire bats observed here (11.6 km yr$^{-1}$) and elsewhere (16.1–37.1 km yr$^{-1}$ in Peru and Argentina) made detection of the same lineage in both continents unlikely during our observational window [17,18,24]. Interestingly, however, three out of five viral lineages circulated only in northern or southern Costa Rica, suggesting they went extinct before traversing the country (figure 3*b*). This raises the possibility that the reliance of VBRV on sufficiently sized bat metapopulations for long-term maintenance

combined with the narrow geography of Central America increases the probability of viral extinction [20]. Consistent with this hypothesis, viral lineages in mainland South America sustain transmission over larger geographical areas, whereas outbreaks on the island of Trinidad fail to persist and are driven by introductions from the mainland, analogous to our observations [18,47]. Ultimately, contemporaneous sequencing of viruses from across the Americas is needed to determine the relative importance of Central America as a corridor or sink of VBRV since existing data are strongly biased towards North and South America. These data could also determine the true frequency of international viral dispersal, which may be under-estimated here due to the limited availability of viral sequences from Central America. Ideally, these datasets could adopt viral whole-genome sequencing, which is expected to increase the resolution of spatio-temporal inferences compared to the single-gene approach used here [48].

The pattern of recurrent lineage invasion and replacement we observed provides new insights into the epidemiological mechanisms that underlie rabies endemicity. The ability of successive viral introductions to invade despite the recent presence of other viruses is unlikely to be explained by anti-genic differences that allow viruses to evade host immunity as rabies antibodies are broadly protective within and among divergent lyssavirus species [49]. Rapid replenishment of susceptibles through births is also unlikely, given the slow reproductive rates of vampire bats (1 offspring per female per year) [43]. Instead, our findings are more likely to be explained by a low $R_0$ of rabies in bats, which creates frequent stochastic extinctions that leave large fractions bat populations suscep-tible for later viral invasions [20]. The low frequency of vampire bat dispersal may also create fine-scale spatial hetero-geneity in susceptibility, wherein colonies that evade one invasion remain susceptible to later viral introductions. Similar extinction–reinvasion dynamics have been observed at fine spatial scales in carnivore rabies. For example, endemicity of rabies in domestic dogs in Bangui, a city in Central African Republic, arises from frequent human-mediated re-introduc-tion of infected dogs that trigger short-lived chains of transmission [50]. However, in other rabies systems, including dog rabies in Tanzania and raccoon rabies in North America,

viral lineages appear pervasively endemic and may even co-circulate [48,51]. These differences are likely to reflect both spatial heterogeneity in rabies persistence (e.g. source–sink dynamics) and observational artefacts, with all rabies viruses experiencing extinctions at some spatial scale. How host popu-lation dynamics, behaviour and environmental context interact to define the scale of viral maintenance remains an outstanding question across rabies systems.

In conclusion, this study shows that combining phyloge-netic inference of relatively small amounts of longitudinal pathogen sequence data with statistical analyses of surveillance data can disentangle endemic pathogen maintenance from recurrent epidemic fadeouts and spatial re-introductions. For VBRV, this information reveals previously unrecognized spatial structure, clarifies the mechanisms underlying ecologi-cal or environmental drivers of cross-species transmission and provides new opportunities for prevention and control. The increasing availability of molecular sequence data should make similar approaches practical in other wildlife disease sys-tems where pathogens persist through spatio-temporal processes, but the dynamics of spatial spread are unobservable from ecological data.

Data accessibility. Sequences generated through this study have been deposited into GenBank (accession numbers no. MK766719–MK766741). Sequence alignments can be extracted from BEAST xml files, which are provided in the electronic supplementary material. Metadata for spatio-temporal analyses are provided as an electronic supplementary material, text file. Code for spatio-temporal statistical analyses is provided as an R script.

Authors' contributions. D.G.S. conducted the statistical and phylogenetic analyses and wrote the first draft of the manuscript; S.L.F.G., G.L. and R.G.B. carried out the molecular laboratory work; B.L. coordi-nated the study, contributed to the laboratory and statistical analyses, and critically revised the manuscript. All authors gave final approval for publication and agree to be held accountable for the work performed therein.

Competing interests. We declare we have no competing interests.

Funding. This study was funded by SENASA-Costa Rica. D.G.S. was supported by a Sir Henry Dale Fellowship, jointly funded by the Wellcome Trust and Royal Society (grant no. 102507/Z/13/Z).

Acknowledgements. We thank members of the Streicker lab, Mafalda Viana and four anonymous reviewers for suggestions on previous versions of this manuscript.

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
