## [Reviewer comments · Proceedings of the Royal Society B: Biological Sciences]

Review History

RSPB-2019-0840.R0 (Original submission)

Review form: Reviewer 1

Recommendation

Accept with minor revision (please list in comments)

Scientific importance: Is the manuscript an original and important contribution to its field?

Excellent

General interest: Is the paper of sufficient general interest?

Good

Quality of the paper: Is the overall quality of the paper suitable?

Good

Is the length of the paper justified?

Yes

Should the paper be seen by a specialist statistical reviewer?

No

Do you have any concerns about statistical analyses in this paper? If so, please specify them explicitly in your report.

No

It is a condition of publication that authors make their supporting data, code and materials available - either as supplementary material or hosted in an external repository. Please rate, if applicable, the supporting data on the following criteria.

Is it accessible?

Yes

Is it clear?

Yes

Is it adequate?

Yes

Do you have any ethical concerns with this paper?

No

Comments to the Author

The authors use genetic information on vampire bat rabies to make inferences about the colonization/extinction dynamics in Costa Rica. They concluded that there have been multiple viral dispersals across the region and that the apparent endemicity of the virus is more likely to be multiple lineages invading and dying out. I thought this was a very interesting paper addressing an important topic. There are few issues that I thought could be improved.

1. Currently the paper is very focused on VBRV. I thought the discussion and introduction could also touch on how this work relates to other rabies systems as well as more generally to the persistence of acute diseases in wildlife systems. In addition, the authors could consider some additional text on how researchers may under/overestimate R_0 , critical community size (or area in this case), and other metrics by lumping across distinct lineages.

2. Management application (ln 299-304): the authors note that the international viral dispersal suggests that Central America may focus on narrow international borders. I think this should be followed by some comment about the dispersal distances of vampire bats as that would determine the width of such an application. I also believe the authors results suggest that VRBV is circulating at very large scales such that management actions may need to be coordinated across country boundaries. Currently the authors only suggest data sharing and surveillance systems across boundaries.

3. Putative extinctions: Comparing the detection intervals to assess the possibility of extinction of certain lineages would seem to assume some consistency in the sampling intensity over time. Given the obvious nature of rabies deaths I would not expect this to be an issue, but it is maybe worth some additional discussion from the authors.

3b. In addition, some of the context of rabies in Costa Rica is not stated in the manuscript. For

example, the number of cattle cases/outbreaks per year is unknown or not describe, but based on the description in Fig. 3 (ln 494) it seems like there may be ?many? outbreaks but no isolates. If true, then a reader may question whether the lineages are absent or unsampled. An alternative to the interval model would be to perhaps do a multinomial regression for lineage type as a function of time where the null model is that they are all equally likely to be chosen given an isolate was sampled through time. Ideally, the probability of choosing an 'extinct' lineage should be estimated to be zero with relatively tight confidence intervals. For the interval model it may be worth stating that the intervals are truncated by the last isolate date and so are a minimum estimate.

4. I thought the modeling of the different lineages as different data partitions, but with common parameters was nicely done given the sample sizes available. No change requested.

Minor fixes/comments:

ln 69. Specify if this cost is globally or for a specific country.

ln 262. Costa Rica as an 'epidemiological sink'. It is not clear from the data presented that Costa Rica differs from other countries in the persistence of VBRV, which is what seems to be implied by using the sink terminology.

ln 185. Not clear here what you mean by "given the observation history of each lineage". To be more transparent and repeatable, including a sup. info. with the data and code would help. Similarly for ln 498 you referring to the 95% CI of what estimate? I'm guessing the mean. If so, then I think only the blue line is needed since you are asking whether putative extinctions fall outside the expected distribution rather than the estimate of the mean.

Fig. 1. Double check abbreviations. Both GUA and GUY are used but GUA is not given a country name. This is an issue in the figure and the legend.

Review form: Reviewer 2

Recommendation

Major revision is needed (please make suggestions in comments)

Scientific importance: Is the manuscript an original and important contribution to its field?

Good

General interest: Is the paper of sufficient general interest?

Good

Quality of the paper: Is the overall quality of the paper suitable?

Acceptable

Is the length of the paper justified?

Yes

Should the paper be seen by a specialist statistical reviewer?

Yes

Do you have any concerns about statistical analyses in this paper? If so, please specify them explicitly in your report.

Yes

It is a condition of publication that authors make their supporting data, code and materials available - either as supplementary material or hosted in an external repository. Please rate, if applicable, the supporting data on the following criteria.

Is it accessible?

Yes

Is it clear?

Yes

Is it adequate?

Yes

Do you have any ethical concerns with this paper?

No

Comments to the Author

The paper of Streicker et al describes spatiotemporal patterns of bat rabies in Costa Rica since 1985. The data set is valuable and supplies opportunity to understand vampire bat rabies from views of phylodynamics.

Major concern:

Several important procedures and key parameters that may affect final result are missing in the current ms. All the results and conclusions should be made based on correct model setting (by BEAST).

Re the mutation rate, the author used $5.6E-4$ as mean value for N gene in their xml file. However, Troupin et al PLOS Pathogens 2016, and Tian et al PLOS Pathogens 2018, in previous studies, have provided an estimation of roughly $2E-4$ for N gene based on larger data set. This change will of course influence all estimations about time (year) in this paper.

The coalescent model (skyline) used should be provided in the main text. Besides, how and why choose this model should be provided as well.

Decision letter (RSPB-2019-0840.R0)

10-Jun-2019

Dear Dr Streicker:

I am writing to inform you that your manuscript RSPB-2019-0840 entitled "Phylodynamics reveals extinction-recolonization dynamics underpin apparently endemic vampire bat rabies in Costa Rica" has, in its current form, been rejected for publication in Proceedings B.

This action has been taken on the advice of referees and the Associate Editor, who have recommended that important revisions are necessary. With this in mind we would be happy to

consider a resubmission, provided the comments of the referees are fully addressed. However please note that this is not a provisional acceptance.

Sincerely,

Professor Hans Heesterbeek
mailto:proceedingsb@royalsociety.org

Associate Editor
Board Member: 1
Comments to Author:

Reviewers were positive about your manuscript "Phylodynamics reveals extinction-recolonization dynamics underpin apparently endemic vampire bat rabies in Costa Rica". It includes an important data set and draws interesting conclusions about endemicity and dispersal of rabies virus in central America.

However, the reviewers had a some concerns that should be addressed. Most importantly, some details of the analyses which would be required to fully assess the statistical aspects of the paper were not clearly communicated. While the parameters are included in the supplemental BEAST XML files, they should be described and justified in the manuscript. It would be valuable to address how uncertainty in key parameters of the model, such as the mutation rate and potential regions with missing data, affects the overall conclusions of the study.

The paper also could be improved by making connections to other rabies systems and wildlife disease.

Reviewer(s)' Comments to Author:

Referee: 1

Comments to the Author(s)

The authors use genetic information on vampire bat rabies to make inferences about the colonization/extinction dynamics in Costa Rica. They concluded that there have been multiple viral dispersals across the region and that the apparent endemicity of the virus is more likely to be multiple lineages invading and dying out. I thought this was a very interesting paper addressing an important topic. There are few issues that I thought could be improved.

1. Currently the paper is very focused on VBRV. I thought the discussion and introduction could also touch on how this work relates to other rabies systems as well as more generally to the persistence of acute diseases in wildlife systems. In addition, the authors could consider some additional text on how researchers may under/overestimate R_0 , critical community size (or area in this case), and other metrics by lumping across distinct lineages.

2. Management application (ln 299-304): the authors note that the international viral dispersal suggests that Central America may focus on narrow international borders. I think this should be followed by some comment about the dispersal distances of vampire bats as that would determine the width of such an application. I also believe the authors results suggest that VBRV is circulating at very large scales such that management actions may need to be coordinated across country boundaries. Currently the authors only suggest data sharing and surveillance systems across boundaries.

3. Putative extinctions: Comparing the detection intervals to assess the possibility of extinction of certain lineages would seem to assume some consistency in the sampling intensity over time. Given the obvious nature of rabies deaths I would not expect this to be an issue, but it is maybe worth some additional discussion from the authors.

3b. In addition, some of the context of rabies in Costa Rica is not stated in the manuscript. For example, the number of cattle cases/outbreaks per year is unknown or not describe, but based on the description in Fig. 3 (ln 494) it seems like there may be many outbreaks but no isolates. If true, then a reader may question whether the lineages are absent or unsampled. An alternative to the interval model would be to perhaps do a multinomial regression for lineage type as a function of time where the null model is that they are all equally likely to be chosen given an isolate was sampled through time. Ideally, the probability of choosing an 'extinct' lineage should be estimated to be zero with relatively tight confidence intervals. For the interval model it may be worth stating that the intervals are truncated by the last isolate date and so are a minimum estimate.

4. I thought the modeling of the different lineages as different data partitions, but with common parameters was nicely done given the sample sizes available. No change requested.

Minor fixes/comments:

ln 69. Specify if this cost is globally or for a specific country.

ln 262. Costa Rica as an 'epidemiological sink'. It is not clear from the data presented that Costa Rica differs from other countries in the persistence of VBRV, which is what seems to be implied by using the sink terminology.

ln 185. Not clear here what you mean by "given the observation history of each lineage". To be more transparent and repeatable, including a sup. info. with the data and code would help.

Similarly for In 498 you referring to the 95% CI of what estimate? I'm guessing the mean. If so, then I think only the blue line is needed since you are asking whether putative extinctions fall outside the expected distribution rather than the estimate of the mean.

Fig. 1. Double check abbreviations. Both GUA and GUY are used but GUA is not given a country name. This is an issue in the figure and the legend.

Referee: 2

Comments to the Author(s)

The paper of Streicker et al describes spatiotemporal patterns of bat rabies in Costa Rica since 1985. The data set is valuable and supplies opportunity to understand vampire bat rabies from views of phylodynamics.

Major concern:

Serval important procedures and key parameters that may affect final result are missing in the current ms. All the results and conclusions should be made based on correct model setting (by BEAST).

Re the mutation rate, the author used $5.6E-4$ as mean value for N gene in their xml file. However, Troupin et al PLOS Pathogens 2016, and Tian et al PLOS Pathogens 2018, in previous studies, have provided an estimation of roughly $2E-4$ for N gene based on larger data set. This change will of course influence all estimations about time (year) in this paper.

The coalescent model (skyline) used should be provided in the main text. Besides, how and why choose this model should be provided as well.

Author's Response to Decision Letter for (RSPB-2019-0840.R0)

See Appendix A.

RSPB-2019-1527.R0

Review form: Reviewer 1

Recommendation

Accept as is

Scientific importance: Is the manuscript an original and important contribution to its field?

Good

General interest: Is the paper of sufficient general interest?

Good

Quality of the paper: Is the overall quality of the paper suitable?

Good

Is the length of the paper justified?

Yes

Should the paper be seen by a specialist statistical reviewer?

No

Do you have any concerns about statistical analyses in this paper? If so, please specify them explicitly in your report.

No

It is a condition of publication that authors make their supporting data, code and materials available - either as supplementary material or hosted in an external repository. Please rate, if applicable, the supporting data on the following criteria.

Is it accessible?

Yes

Is it clear?

Yes

Is it adequate?

Yes

Do you have any ethical concerns with this paper?

No

Comments to the Author

The authors seem to have addressed my previous recommendations and I have no further suggestions.

Review form: Reviewer 3

Recommendation

Accept with minor revision (please list in comments)

Quality of the paper: Is the overall quality of the paper suitable?

A good paper worth publishing in Proc. R. Soc. B

Is the length of the paper justified?

Yes

Do you have any ethical concerns with this paper?

No

Comments to the Author

Streicker et al. here present a revised version of their study titled 'Phylogenetics reveals extinction-recolonization dynamics underpin apparently endemic vampire bat rabies in Costa

Rica'. I precise that I was not a reviewer of the initial submission. I found the study interesting, as well as the manuscript well written and relatively clear to follow. I also agree with the previous comment of another reviewer on the fact that the phylogeographic workflow makes sense (i.e. first identifying the different lineages with a discrete approach and then performing distinct continuous phylogeographic analyses on each lineage while sharing some overall models).

However, I have several concerns:

- 1.201-215 ('Temporal analysis of putative national viral invasions and extinctions'): I found this paragraph of the Methods section extremely difficult to follow. Was this analysis based on ancestral nodes occurrence? On how many posterior trees did you base this analysis? Since this is the methodological novelty of the present study, I believe that its description should be more detailed. For instance, the mathematical expression of the regression should be explicitly written. A figure detailing/presenting the approach could also help explaining exactly what was performed.

- sampling size: the number of sequences originated from Costa Rica is rather low. The authors should at least discuss the statistical power of their new approach in light of the sampling size.

- 1.176 (discrete phylogeographic analysis): why only including 35 sequences originated from other countries? This is a rather low proportion of non-Costa Rican sequences for the 'international' analysis. Were you limited by the sequences available on GenBank? (1.227-232:) the potential impact of sampling bias is never discussed nor mentioned. Given the low proportion of sequences originated from other countries, this should be addressed in the text. To what extent the authors can be sure that they did not underestimated the number of distinct lineages circulating in Costa Rica?

- 1.238 (dispersal velocity): 'dispersal velocity' is too vague. Does it make reference to the 'wavefront velocity' or the 'branch dispersal velocity' (please state exactly which statistic you estimated). Also, what about the comparison of this dispersal statistic between the two main clades? The author should also compare and discuss these values with those estimated on other datasets.

- 1.290-292: I am not sure to understand, and my apologies if I missed something, but how exactly are your results supporting this statement - please develop or correct it.

- figure 3: I am not sure to get the real added value of that figure. Also, why not also reporting the inferred position of ancestral nodes? (But this question is directly related to my previous question above about the methodology).

Other minor concern:

- 1.126-127 (and in general): a simple sampling would actually be welcome.

Review form: Reviewer 4

Recommendation

Accept with minor revision (please list in comments)

Quality of the paper: Is the overall quality of the paper suitable?

A very good paper making an important contribution to the field

Is the length of the paper justified?

Yes

Do you have any ethical concerns with this paper?

No

Comments to the Author

The statistical analyses employed here overall seem sound and provide interesting insights into the circulation of an endemic pathogen at regional and national scales. In particular, the use of discrete and continuous phylogeographic methods are clever. My key concern is about the multinomial regression. What were the other classes in the regression (0, 1 and?). Was time treated as a fixed or random effect? Was there evidence for homoscedasticity in the residuals as can be the case for GLMs with time series? Seems like more detail is needed to assess this analysis. Initially, I was concerned that undetected virus circulation in the bats wouldn't be detected and this would be a significant limitation, but the arguments in the discussion satisfied my concerns. In fact, it may be good for the authors to stress this in the methods by adding a sentence or two. Otherwise, most of my comments are relatively minor.

138: Why only one gene? For those unfamiliar with rabies, some explanation of why this small region (1500 bp) was targeted and some of the limitations with just using a small region of the rabies genome is needed.

162: Using separate substitution models for each codon position, to my knowledge, is not readily doable using the BEAUTI interface. How was this done?

170: A sup table with all of these BF and likelihood values is needed here.

173: In this case, why weren't too separate analyses performed?

194: More detail is required here i.e. which particular statistic.

195-6: This detail is superfluous - could be dropped.

196: Why 6 months?

198: Why weren't ESS's checked? This seems like an obvious omission.

220: Just for the N gene correct?

Fig. 1c. Where is CR? Am I missing something?

Decision letter (RSPB-2019-1527.R0)

30-Jul-2019

Dear Dr Streicker:

Your manuscript has now been peer reviewed and the reviews have been assessed by an Associate Editor. The reviewers' comments (not including confidential comments to the Editor) and the comments from the Associate Editor are included at the end of this email for your reference. As you will see, the reviewers and the Editors have raised some concerns with your manuscript and we would like to invite you to revise your manuscript to address them.

When submitting your revision please upload a file under "Response to Referees" in the "File

Upload" section. This should document, point by point, how you have responded to the reviewers' and Editors' comments, and the adjustments you have made to the manuscript. We require a copy of the manuscript with revisions made since the previous version marked as 'tracked changes' to be included in the 'response to referees' document.

Research ethics:

Use of animals and field studies:

Please submit a copy of your revised paper within three weeks. If we do not hear from you within this time your manuscript will be rejected. If you are unable to meet this deadline please let us know as soon as possible, as we may be able to grant a short extension.

Best wishes,

Professor Hans Heesterbeek
mailto:proceedingsb@royalsociety.org

Associate Editor

Comments to Author:

Dear authors,

Thank you for your revised submission of your manuscript "Phylogenetics reveals extinction-recolonization dynamics underpin apparently endemic vampire bat rabies in Costa Rica". It is an interesting paper with the potential to have an important impact. Unfortunately, the key concern from the previous submission has not been resolved. Specifically we stated "Most importantly, some details of the analyses which would be required to fully assess the statistical aspects of the paper were not clearly communicated."

Both reviewers 2 and 3 specified that they lacked methodological details required to fully assess this revised submission. Therefore, we cannot consider it for publication in Proc B at this time.

Reviewer(s)' Comments to Author:

Referee: 1

Comments to the Author(s).

The authors seem to have addressed my previous recommendations and I have no further suggestions.

Referee: 3

Comments to the Author(s).

Streicker et al. here present a revised version of their study titled 'Phylogenetics reveals extinction-recolonization dynamics underpin apparently endemic vampire bat rabies in Costa Rica'. I precise that I was not a reviewer of the initial submission. I found the study interesting, as well as the manuscript well written and relatively clear to follow. I also agree with the previous comment of another reviewer on the fact that the phylogeographic workflow makes sense (i.e. first identifying the different lineages with a discrete approach and then performing distinct continuous phylogeographic analyses on each lineage while sharing some overall models). However, I have several concerns:

- 1.201-215 ('Temporal analysis of putative national viral invasions and extinctions'): I found this paragraph of the Methods section extremely difficult to follow. Was this analysis based on

ancestral nodes occurrence? On how many posterior trees did you base this analysis? Since this is the methodological novelty of the present study, I believe that its description should be more detailed. For instance, the mathematical expression of the regression should be explicitly written. A figure detailing/presenting the approach could also help explaining exactly what was performed.

- sampling size: the number of sequences originated from Costa Rica is rather low. The authors should at least discuss the statistical power of their new approach in light of the sampling size.

- 1.176 (discrete phylogeographic analysis): why only including 35 sequences originated from other countries? This is a rather low proportion of non-Costa Rican sequences for the 'international' analysis. Were you limited by the sequences available on GenBank? (1.227-232:) the potential impact of sampling bias is never discussed nor mentioned. Given the low proportion of sequences originated from other countries, this should be addressed in the text. To what extent the authors can be sure that they did not underestimated the number of distinct lineages circulating in Costa Rica?

- 1.238 (dispersal velocity): 'dispersal velocity' is too vague. Does it make reference to the 'wavefront velocity' or the 'branch dispersal velocity' (please state exactly which statistic you estimated). Also, what about the comparison of this dispersal statistic between the two main clades? The author should also compare and discuss these values with those estimated on other datasets.

- 1.290-292: I am not sure to understand, and my apologies if I missed something, but how exactly are your results supporting this statement - please develop or correct it.

- figure 3: I am not sure to get the real added value of that figure. Also, why not also reporting the inferred position of ancestral nodes? (But this question is directly related to my previous question above about the methodology).

Other minor concern:

- 1.126-127 (and in general): a simple sampling would actually be welcome.

Referee: 4

Comments to the Author(s).

The statistical analyses employed here overall seem sound and provide interesting insights into the circulation of an endemic pathogen at regional and national scales. In particular, the use of discrete and continuous phylogeographic methods are clever. My key concern is about the multinomial regression. What were the other classes in the regression (0, 1 and?). Was time treated as a fixed or random effect? Was there evidence for homoscedasticity in the residuals as can be the case for GLMs with time series? Seems like more detail is needed to assess this analysis. Initially, I was concerned that undetected virus circulation in the bats wouldn't be detected and this would be a significant limitation, but the arguments in the discussion satisfied my concerns. In fact, it may be good for the authors to stress this in the methods by adding a sentence or two. Otherwise, most of my comments are relatively minor.

138: Why only one gene? For those unfamiliar with rabies, some explanation of why this small region (1500 bp) was targeted and some of the limitations with just using a small region of the rabies genome is needed.

162: Using separate substitution models for each codon position, to my knowledge, is not readily doable using the BEAUTI interface. How was this done?

170: A sup table with all of these BF and likelihood values is needed here.

173: In this case, why weren't too separate analyses performed?

194: More detail is required here i.e. which particular statistic.

195-6: This detail is superfluous - could be dropped.

196: Why 6 months?

198: Why weren't ESS's checked? This seems like an obvious omission.

220: Just for the N gene correct?

Fig. 1c. Where is CR? Am I missing something?

Author's Response to Decision Letter for (RSPB-2019-1527.R0)

See Appendix B.

RSPB-2019-1527.R1 (Revision)

Review form: Reviewer 3

Recommendation

Accept with minor revision (please list in comments)

Scientific importance: Is the manuscript an original and important contribution to its field?

Good

General interest: Is the paper of sufficient general interest?

Good

Quality of the paper: Is the overall quality of the paper suitable?

Good

Is the length of the paper justified?

Yes

Should the paper be seen by a specialist statistical reviewer?

No

Do you have any concerns about statistical analyses in this paper? If so, please specify them explicitly in your report.

No

It is a condition of publication that authors make their supporting data, code and materials available - either as supplementary material or hosted in an external repository. Please rate, if applicable, the supporting data on the following criteria.

Is it accessible?

Yes

Is it clear?

Yes

Is it adequate?

Yes

Do you have any ethical concerns with this paper?

No

Comments to the Author

Going through the revised manuscript, I realise that there were indeed specific methodological aspects that I misunderstood in study. While I think the text is clearer now, I also admit my lack of expertise to properly assess the novel analysis presented here (temporal analysis of putative national viral invasions and extinctions). However, I get the general idea and really appreciate reading this study joining phylodynamic (involving a relevant combination of discrete and continuous phylogeographic inferences) and modelling approaches. My only remaining general comment is that the link between the two (even if stated e.g. in line 222) should be even more explicit or put forward elsewhere (abstract?). Here are a series of other minor comments:

- figure S1: I would actually use a continuous color scale to color the dots. Also, how do you refer/indicate the samples that were unavailable for sequencing? This is not obvious to me right now and I think that sequenced cases should be clearly highlighted on the map. And finally, another detail: what's the background map (I guess elevation...)? Please report its nature, a color scale and a source. This figure is actually important regarding the subsequent analyses performed on these outbreak records. This is of course only a matter of personal preference, but I would prefer such a modified version of Figure 1 rather than the current version of Figure 3.

- figure 1: I really like it, but I would also really advise the authors to better highlight the sequences coming from Costa Rica (switching colors? coloring sequence names?). This should be directly obvious for the reader where these sequences are in the tree...

- figure 2: could add some kind of temporal information on it? This could potentially better illustrate spread directions.

- abstract: it remains not obvious to me what the authors exactly mean by "no lineage persisted was detected throughout the duration of surveillance".

Review form: Reviewer 4 (Nicholas Fountain-Jones)

Recommendation

Accept with minor revision (please list in comments)

Scientific importance: Is the manuscript an original and important contribution to its field?

Good

General interest: Is the paper of sufficient general interest?

Good

Quality of the paper: Is the overall quality of the paper suitable?

Good

Is the length of the paper justified?

Yes

Should the paper be seen by a specialist statistical reviewer?

Yes

Do you have any concerns about statistical analyses in this paper? If so, please specify them explicitly in your report.

No

It is a condition of publication that authors make their supporting data, code and materials available - either as supplementary material or hosted in an external repository. Please rate, if applicable, the supporting data on the following criteria.

Is it accessible?

Yes

Is it clear?

Yes

Is it adequate?

Yes

Do you have any ethical concerns with this paper?

No

Comments to the Author

The authors have done a good job of addressing my statistical concerns.

My only suggestion is to include a reference to Figure 4 line 271 to make it clear where the CIs are they are referring to.

Decision letter (RSPB-2019-1527.R1)

12-Sep-2019

Dear Dr Streicker

I am pleased to inform you that your manuscript RSPB-2019-1527.R1 entitled "Phylodynamics reveals extinction-recolonization dynamics underpin apparently endemic vampire bat rabies in Costa Rica" has been accepted for publication in Proceedings B.

The referees have recommended publication, but also request some minor revisions to your manuscript. Therefore, I invite you to respond to the referees' comments and revise your manuscript. Because the schedule for publication is very tight, it is a condition of publication that you submit the revised version of your manuscript within 7 days. If you do not think you will be able to meet this date please let us know.

To revise your manuscript, log into <https://mc.manuscriptcentral.com/prsb> and enter your Author Centre, where you will find your manuscript title listed under "Manuscripts with Decisions." Under "Actions," click on "Create a Revision." Your manuscript number has been appended to denote a revision. You will be unable to make your revisions on the originally

submitted version of the manuscript. Instead, revise your manuscript and upload a new version through your Author Centre.

[http://datadryad.org/submit?journalID=RSPB&manu=\(Document not available\)](http://datadryad.org/submit?journalID=RSPB&manu=(Document+not+available)) which will take you to your unique entry in the Dryad repository. If you have already submitted your data to dryad you can make any necessary revisions to your dataset by following the above link. Please see <https://royalsociety.org/journals/ethics-policies/data-sharing-mining/> for more details.

Sincerely,

Professor Hans Heesterbeek
Editor, Proceedings B
<mailto:proceedingsb@royalsociety.org>

Associate Editor:

Comments to Author:

We are pleased to accept your manuscript "Phylodynamics reveals extinction-recolonization dynamics underpin apparently endemic vampire bat rabies in Costa Rica" for publication following minor revisions. Both reviewers make some comments on the figures that could increase readability and clarity, as well as some suggestions to improve the text, which we would like the authors to incorporate before publication. Thank you for your submission and we look forward to receiving your revised manuscript.

Reviewer(s)' Comments to Author:

Referee: 3

Comments to the Author(s)

Going through the revised manuscript, I realise that there were indeed specific methodological aspects that I misunderstood in study. While I think the text is clearer now, I also admit my lack of expertise to properly assess the novel analysis presented here (temporal analysis of putative national viral invasions and extinctions). However, I get the general idea and really appreciate reading this study joining phylodynamic (involving a relevant combination of discrete and continuous phylogeographic inferences) and modelling approaches. My only remaining general comment is that the link between the two (even if stated e.g. in line 222) should be even more explicit or put forward elsewhere (abstract?). Here are a series of other minor comments:

- figure S1: I would actually use a continuous color scale to color the dots. Also, how do you refer/indicate the samples that were unavailable for sequencing? This is not obvious to me right now and I think that sequenced cases should be clearly highlighted on the map. And finally, another detail: what's the background map (I guess elevation...)? Please report its nature, a color scale and a source. This figure is actually important regarding the subsequent analyses performed on these outbreak records. This is of course only a matter of personal preference, but I would prefer such a modified version of Figure 1 rather than the current version of Figure 3.

- figure 1: I really like it, but I would also really advise the authors to better highlight the sequences coming from Costa Rica (switching colors? coloring sequence names?). This should be directly obvious for the reader where these sequences are in the tree...

- figure 2: could add some kind of temporal information on it? This could potentially better illustrate spread directions.

- abstract: it remains not obvious to me what the authors exactly mean by "no lineage persisted was detected throughout the duration of surveillance".

Referee: 4

Comments to the Author(s)

The authors have done a good job of addressing my statistical concerns.

My only suggestion is to include a reference to Figure 4 line 271 to make it clear where the CIs are they are referring to.

Author's Response to Decision Letter for (RSPB-2019-1527.R1)

See Appendix C.

Decision letter (RSPB-2019-1527.R2)

17-Sep-2019

Dear Dr Streicker

I am pleased to inform you that your manuscript entitled "Phylodynamics reveals extinction-recolonization dynamics underpin apparently endemic vampire bat rabies in Costa Rica" has been accepted for publication in Proceedings B.

Open Access

Paper charges

Sincerely,

Proceedings B

Appendix A

University
of Glasgow

Institute of Biodiversity,
Animal Health & Comparative Medicine

CVR
Medical Research Council
University of Glasgow
Centre for Virus Research

Dr. Daniel Streicker

Wellcome Trust & Royal Society Sir Henry Dale Fellow
Institute of Biodiversity, Animal Health and Comparative Medicine
MRC-University of Glasgow Centre for Virus Research
University of Glasgow
Glasgow, G12 8QQ
Tel: +44 (0)1413306632
Email: daniel.streicker@glasgow.ac.uk

28 June 2019

Dear Editors of *Proceedings of the Royal Society – Biological Sciences*,

We are pleased to submit a revised version of our manuscript “*Phylodynamics reveals extinction-recolonization dynamics underpin apparently endemic vampire bat rabies in Costa Rica*” for consideration for publication as a Research Article in *Proceedings B*.

My co-authors and I appreciated the constructive nature of the suggestions from the reviewers. We have addressed each of these concerns and now present a revised manuscript which we believe is both stronger and clearer. In particular, we have provided greater context relating our findings to other infectious disease systems, including rabies viruses associated with other reservoir hosts, but also other human, domestic animal, and wildlife diseases. We believe these changes expand the reach of the paper to a broader readership across infectious disease ecology and evolution, public health, and wildlife management. In addition, we provided greater explanation of our phylogenetic analyses, paying particular attention to explaining our choice of priors. We did not follow the suggestion of Reviewer 2 to fix the substitution rate at a lower value than that which was estimated from our data, and our explanation for this decision is detailed below. Finally, we conducted additional statistical analyses following suggestions from Reviewer 1. In all cases, these analyses strengthened the conclusions from the previous version of the manuscript. The code and data to reproduce our statistical analyses is also now provided.

In the attached document, we present a point-by-point response to each reviewers’ critique, with our responses in **bold** typeface.

Thank you for your continued consideration.

With best regards,

Daniel Streicker and Bernal Leon
(On behalf of all authors)

Point-by-point responses to Reviewer's Critiques

Referee: 1

Comments to the Author(s)

The authors use genetic information on vampire bat rabies to make inferences about the colonization/extinction dynamics in Costa Rica. They concluded that there have been multiple viral dispersals across the region and that the apparent endemicity of the virus is more likely to be multiple lineages invading and dying out. I thought this was a very interesting paper addressing an important topic. There are few issues that I thought could be improved.

1. Currently the paper is very focused on VBRV. I thought the discussion and introduction could also touch on how this work relates to other rabies systems as well as more generally to the persistence of acute diseases in wildlife systems. In addition, the authors could consider some additional text on how researchers may under/overestimate R_0 , critical community size (or area in this case), and other metrics by lumping across distinct lineages.

We thank the reviewer for this suggestion. The best published examples of lineage extinction and replacement involve acute human and domestic pathogens (i.e., Dengue and Foot and Mouth Disease Virus) and we have added these illustrative examples to the introduction manuscript along with references (P3L62-64). We suspect analogous dynamics could be widespread in wildlife diseases and have mentioned some examples where spatial dynamics are clearly important for wildlife diseases (P3L65-68) but suggest the scarcity of clear examples of extinction-recolonization dynamics reflects the rarity of longitudinal genotyped/serotype data. As suggested, we have elaborated on how unrecognized cryptic extinction-recolonization dynamics could alter model selection for studies attempting to identify maintenance mechanisms, estimates of R_0 and critical community size (P3L70-73). Following on the new concepts in the introduction, we re-structured and added a new paragraph to the discussion which addresses epidemiological explanations for how the observed dynamics arise and how this relates to other rabies systems (P14L338-359).

2. Management application (In 299-304): the authors note that the international viral dispersal suggests that Central America may focus on narrow international borders. I think this should be followed by some comment about the dispersal distances of vampire bats as that would determine the width of such an application. I also believe the authors results suggest that VRBV is circulating at very large scales such that management actions may need to be coordinated across country boundaries. Currently the authors only suggest data sharing and surveillance systems across boundaries.

We added a statement and a reference on the dispersal behaviours of vampire bats (P13L309-313). Regarding international data sharing, we agree with the reviewer and were trying to make this same point in the original version of the manuscript, but it may have been unclear. We changed the sentence in question to explicitly encourage data sharing across all of Latin America, rather than only at countries bordering Costa Rica (P13L313-315).

3. Putative extinctions: Comparing the detection intervals to assess the possibility of extinction of certain lineages would seem to assume some consistency in the sampling

intensity over time. Given the obvious nature of rabies deaths I would not expect this to be an issue, but it is maybe worth some additional discussion from the authors.

Please see our response below, which describes new analyses addressing sampling intensity through time.

3b. In addition, some of the context of rabies in Costa Rica is not stated in the manuscript. For example, the number of cattle cases/outbreaks per year is unknown or not describe, but based on the description in Fig. 3 (ln 494) it seems like there may be ?many? outbreaks but no isolates. If true, then a reader may question whether the lineages are absent or unsampled. An alternative to the interval model would be to perhaps do a multinomial regression for lineage type as a function of time where the null model is that they are all equally likely to be chosen given an isolate was sampled through time. Ideally, the probability of choosing an 'extinct' lineage should be estimated to be zero with relatively tight confidence intervals. For the interval model it may be worth stating that the intervals are truncated by the last isolate date and so are a minimum estimate.

We provided some additional context on our data and the situation of rabies in Costa Rica at the start of the methods section, specifically the number of unsampled outbreaks over the time course of the study, the average number of deaths per outbreak, and and the number of animals killed during these outbreaks (P5L121-132). We also show that the number of viruses sequenced per year was correlated with the number of outbreaks, which reduces the risk that the putative lineage extinctions arose from biased sampling and illustrates that sampling intensity is roughly constant through time.

Nevertheless, we liked the reviewer's suggestion to conduct a multinomial regression model and carried out the suggested analysis. A new figure (Fig. 4) shows the predicted probabilities of each virus being present in Costa Rica over the time period of the study. These results show almost exactly what the reviewer described would be ideal; supporting the earlier suggestions that virus 1b, 1d, and 2 were very likely to have gone extinct (predicted probabilities of presence in 2017 < 0.025), with less certainty for virus 1c (predicted probability of presence = 0.4). These results also contribute new information about the probabilities of viral presence at the beginning of the time series, showing that viruses L1a and L1c were extremely unlikely to have been present during the first half of the time period and invaded between 2010 and 2012. These new analyses are described in the methods (P9L201-203) and results (P10L248-251)

For the interval model, we clarified our text to say that "Values for the time since final detection are therefore conservative lower bounds, *representing minimum estimates.*" (P9L207-209)

4. I thought the modeling of the different lineages as different data partitions, but with common parameters was nicely done given the sample sizes available. No change requested.

Thanks. We agree, this is an underutilized capability of BEAST.

Minor fixes/comments:

In 69. Specify if this cost is globally or for a specific country.

This cost refers to Latin America. This has been clarified.

In 262. Costa Rica as an 'epidemiological sink'. It is not clear from the data presented that Costa Rica differs from other countries in the persistence of VBRV, which is what seems to be implied by using the sink terminology.

This is a fair point and we have expanded our discussion to be explicit that the spatial scale of rabies persistence and whether it always persists through source sink dynamics or metapopulation dynamics remains unclear, both for bat and carnivore hosts (P14L338-359).

Ln 185. Not clear here what you mean by "given the observation history of each lineage". To be more transparent and repeatable, including a sup. info. with the data and code would help. Similarly for Ln 498 you referring to the 95% CI of what estimate? I'm guessing the mean. If so, then I think only the blue line is needed since you are asking whether putative extinctions fall outside the expected distribution rather than the estimate of the mean.

By observation history, we meant the duration of time that the virus went undetected. This has been revised for clarity (P9L214). We also modified the figure as suggested and provided the data and R code for the spatiotemporal analyses.

Fig. 1. Double check abbreviations. Both GUA and GUY are used but GUA is not given a country name. This is an issue in the figure and the legend.

This has been corrected by adding a complete description of country abbreviations in panel 1A. Thanks for catching this oversight.

Referee: 2

Comments to the Author(s)

The paper of Streicker et al describes spatiotemporal patterns of bat rabies in Costa Rica since 1985. The data set is valuable and supplies opportunity to understand vampire bat rabies from views of phylodynamics.

Major concern:

Several important procedures and key parameters that may affect final result are missing in the current ms. All the results and conclusions should be made based on correct model setting (by BEAST).

We thank the reviewer for pointing out some ambiguity in our description of our models. As discussed in more detail below, we have better justified our choices and have added some results to the main text.

Re the mutation rate, the author used $5.6E-4$ as mean value for N gene in their xml file. However, Troupin et al PLOS Pathogens 2016, and Tian et al PLOS Pathogens 2018, in previous studies, have provided an estimation of roughly $2E-4$ for N gene based on larger data set. This change will of course influence all estimations about time (year) in this paper.

The prior on substitution rate for the analysis of the international dataset (the xml file the Reviewer refers to) was based on our preliminary analysis in Tempest and spanned a wide range of possibilities from 0 to $1e-3$ substitutions per site per year, and therefore included those slower rates suggested by other papers. Therefore, our prior did not preclude higher or lower estimates of the substitution rate, had these

been supported by the temporal signals present in the data. The reviewer is correct that our posterior estimate ($4.86e-4$, 95% HPD = $3.54-6.34e-4$) was higher than the global estimate of Troupin et al.. However that estimate is not easily compared to ours for two primary reasons. First, their estimate was for carnivore rabies, not bat rabies. Second, their global estimate was averaged across many independent virus lineages which circulate in different host species and had variable substitution rates, some of which were similar to our estimate. For example mongoose rabies (AF3 virus) and ferret badger rabies both evolved faster than our estimate of vampire bat rabies. The estimate of Tian also was for dog rabies, and is lower than ours, but again, this is expected given the variation in evolutionary rates across carnivore hosts. We have previously shown that rabies is evolutionarily and epidemiologically independent in each bat species (Streicker et al. 2010 Science) and, crucially, that viruses also evolve at dramatically different rates in different bat species, with faster evolution in tropical/subtropical bats such as vampire bats compared to bats in temperate zones (Streicker et al. 2012 PLoS Pathogens). It is therefore more appropriate to design priors on estimates derived from viral lineages maintained by single reservoir host species (rather than larger, but more variable datasets which comprise many different viruses that evolve at different rates) and let the data determine the posterior estimate, as we have done. Encouragingly, our estimate is consistent with previous analyses of two much larger lineages of vampire bat rabies in Peru ($5.49e-4$ and $5.05e-4$; Streicker et al. 2016 PNAS). Based on this, we did not revise date estimates but added the estimated substitution rate (P9L219-221) and provided more justification of our selection of priors on substitution rate in the manuscript (P7L167-175).

The coalescent model (skyline) used should be provided in the main text. Besides, how and why choose this model should be provided as well.

We apologize for this oversight in the description of our methods. Given the potentially complex demographic history in the international dataset, we specified the Bayesian skyline model as a flexible demographic prior that made minimal assumptions. This is now described (P8L178-181) For the continuous phylogeographic models of L1a and L2, we opted for a simpler demographic model of constant population size given that each lineage had small sample size which would have precluded more complex models (see P8L190-191. Importantly however, that analysis is primarily focused on the ancestral reconstruction of the geographic origins of each virus (Fig 2), which is insensitive to the demographic model chosen.

Appendix B

Point-by-point responses to the editor's and reviewers' critiques

Thank you for your revised submission of your manuscript "Phylogenetics reveals extinction-recolonization dynamics underpin apparently endemic vampire bat rabies in Costa Rica". It is an interesting paper with the potential to have an important impact. Unfortunately, the key concern from the previous submission has not been resolved. Specifically we stated "Most importantly, some details of the analyses which would be required to fully assess the statistical aspects of the paper were not clearly communicated."

Both reviewers 2 and 3 specified that they lacked methodological details required to fully assess this revised submission. Therefore, we cannot consider it for publication in Proc B at this time.

Our response: We provided the additional methodological details requested on the original analyses and further explanation of the statistical analysis that we performed in response to the first round of reviews.

Reviewer(s)' Comments to Author:

Referee: 1

Comments to the Author(s).

The authors seem to have addressed my previous recommendations and I have no further suggestions.

Our response: We are again grateful for the earlier suggestions, which substantially improved the paper.

Referee: 3

Comments to the Author(s).

Streicker et al. here present a revised version of their study titled 'Phylogenetics reveals extinction-recolonization dynamics underpin apparently endemic vampire bat rabies in Costa Rica'. I precise that I was not a reviewer of the initial submission. I found the study interesting, as well as the manuscript well written and relatively clear to follow. I also agree with the previous comment of another reviewer on the fact that the phylogeographic workflow makes sense (i.e. first identifying the different lineages with a discrete approach and then performing distinct continuous phylogeographic analyses on each lineage while sharing some overall models). However, I have several concerns:

- 1.201-215 ('Temporal analysis of putative national viral invasions and extinctions'): I found this paragraph of the Methods section extremely difficult to follow. Was this analysis based on ancestral nodes occurrence? On how many posterior trees did you base this analysis? Since this is the methodological novelty of the present study, I believe that its description should be more detailed. For instance, the mathematical expression of the regression should be explicitly written. A figure detailing/presenting the approach could also help explaining exactly what was performed.

Our response: We apologize for the confusion related to this analysis. This analysis did not involve ancestral inferences from the phylogenetic analysis. Instead we modelled (1) the observed time series of each viral lineage using a multinomial logistic regression and (2) the observed inter-outbreak periods (in months) while viruses were known to have circulated. In the revised version of the manuscript we re-wrote this section, adding several sentences to provide more background on the multinomial regression, along with an equation (P9, L206-219). Given that this approach does not directly involve molecular data as the reviewer suspected and that multinomial regression is commonly used for classification, we think our explanation of the approach is sufficient without a methodological figure.

- sampling size: the number of sequences originated from Costa Rica is rather low. The authors should at least discuss the statistical power of their new approach in light of the sampling size.

Our response: As mentioned above, the multinomial regression did not explicitly use sequence data; however, sample size is important to consider regardless. The uncertainty in our model predictions is evident in the 95% confidence intervals shown in Figure 4A. This shows that while sample size was limited, the model is sufficiently powered to detect viral extinctions (lower 95% confidence interval on the probability of virus presence approaches zero) and invasions (lower 95% rises above zero), which was the primary aim of this analysis. We added a statement explaining the statistical power of our approach to the results (P11, L267-271):

"Despite the relatively small sample sizes underlying the lineage-specific time series, the multinomial regression found an effect of year on which virus was observed ($\chi^2 = 29.6$, $p < 0.001$, McFadden's pseudo $r^2 = 0.26$) and had sufficient power to detect viral invasions and extinctions, evidenced by 95% confidence intervals on the predicted probability of viral presence through time rising above or approaching zero, respectively."

- I.176 (discrete phylogeographic analysis): why only including 35 sequences originated from other countries? This is a rather low proportion of non-Costa Rican sequences for the 'international' analysis. Were you limited by the sequences available on GenBank?

Our response: *Since even the 'international' analysis focused on viral invasions to and dispersals from Costa Rica, sequences that could be reasonably expected to share a common ancestor with viruses from Costa Rica were most appropriate. Therefore, we employed a cutoff based on sequence similarity and included all sequences in Genbank above that threshold. More divergent sequences would inform patterns of viral dispersal among non-focal countries, but this was not the aim of our study. The rationale for our sequence selection has been elaborated in the text (P7, L154-159):*

"International analyses focused on the timescale and geographic patterns of viral dispersals into and out of Costa Rica using discrete phylogeographic analyses. We used a cutoff of 98% similarity to any Costa Rican sequence to identify VBRVs available in Genbank that could plausibly have shared a most recent common ancestor with Costa Rican viruses, but also included additional representative VBRV lineages for reference."

(I.227-232:) the potential impact of sampling bias is never discussed nor mentioned. Given the low proportion of sequences originated from other countries, this should be addressed in the text. To what extent the authors can be sure that they did not underestimated the number of distinct lineages circulating in Costa Rica?

Our response: It is true that there could be more invasions into Costa Rica than we detected and, although we did not use the exact terminology suggested by the reviewer, this was discussed in the previous version of the manuscript at lines 335-337 (now P15, L357-361). Unfortunately, the data that would be required to detect additional invasions do not exist in Genbank. As a reminder, this was the first phylogeographic study of vampire bat rabies in all of Central America, so data gaps from other Central American countries were inevitable. Importantly, splitting the lineages found in Costa Rica by adding more data from other countries (if it existed) would only strengthen the central conclusion of our manuscript that rabies only appears endemic in Costa Rica due to recurrent viral invasions. We also previously stated how the limited data that are currently available from Latin America affect spatially informed management and whether Central America is a sink for rabies invasions. We left these points in the revised manuscript and added statements that further describe the sampling biases in existing datasets (P15, L357-364):

"Ultimately, contemporaneous sequencing of viruses from across the Americas is needed to determine the relative importance of Central America as a corridor or sink of VBRV since existing data are strongly biased towards North and South America. These data could also determine the true frequency of international viral dispersal, which may be underestimated here due to the limited availability of viral sequences from Central America. Ideally, these datasets could adopt viral whole genome sequencing, which is expected to increase the resolution of spatiotemporal inferences compared to the single-gene approach used here [48]."

- I.238 (dispersal velocity): 'dispersal velocity' is too vague. Does it make reference to the 'wavefront velocity' or the 'branch dispersal velocity' (please state exactly which statistic you estimated). Also, what about the comparison of this dispersal statistic between the two main clades? The author should also compare and discuss these values with those estimated on other datasets.

Our response: *We used mean branch velocity and the terminology has been clarified. We also added the estimate for the second lineage, though as discussed in the text, this estimate should be treated with caution given the less precise origins of that virus (P11, L256-260).*

"The inferred ancestor of virus L2 occurred further north in central Costa Rica, but was considerably less precise, suggesting that the longer time period of circulation in Costa Rica largely erased evolutionary signal of viral origins (Fig. 2). This led to higher and more variable velocities which should be interpreted cautiously (median = 56.98 km/year, 95% credible interval = 12.62-2522.36)."

As requested, we also provided more a more detailed comparison in the discussion of how our estimated rates compared to previously published reports (P14, L346-349):

"...the relatively slow progression of rabies expansions in vampire bats observed here (11.6 km/year) and elsewhere (16.1 – 37.1 km/year in Peru and Argentina) made detection of the same lineage in both continents unlikely during our observational window [17,18,24]."

- I.290-292: I am not sure to understand, and my apologies if I missed something, but how exactly are your results supporting this statement - please develop or correct it.

Our response: This is an important point and was perhaps not clear enough. Our conclusion is that if rabies is not consistently present in a locality, then changes in bat demography or immunity alone cannot trigger outbreaks – if the virus is locally absent, then it cannot respond to these hypothetical drivers. Therefore, the main way for these factors to influence transmission to occur would be if they increase viral introductions to an area, which can only occur via dispersal of bats. We developed this argument more fully as requested (P13, L310-316):

“An important consequence of extinction-reinvasion dynamics is that environmental or demographic factors alone cannot directly trigger spillover by increasing bat population density, reducing immunity, or altering feeding behaviour since the virus will often not be locally present to respond to these potential drivers. Increases in transmission might instead be explained by environmentally or anthropogenically-driven increases in bat dispersal that connect VBRV-free to nearby VBRV-infected populations [20].”

- figure 3: I am not sure to get the real added value of that figure. Also, why not also reporting the inferred position of ancestral nodes? (But this question is directly related to my previous question above about the methodology).

Our response: As described above, ancestral locations were not used in our spatiotemporal statistical analyses, so they would distract from the main point of this figure. This is the only figure which shows the location of observed viruses and their temporal patterns, so we prefer to keep it in since it provides an intuitive view of lineage presence/absence/introduction through time. Please note we re-sized this figure to make more efficient use of space. However, we would be willing to move this figure to the supplement at the editors' discretion.

Other minor concern:

- I.126-127 (and in general): a simple sampling would actually be welcome.

Our response: Done as suggested

Referee: 4

Comments to the Author(s).

The statistical analyses employed here overall seem sound and provide interesting insights into the circulation of an endemic pathogen at regional and national scales. In particular, the use of discrete and continuous phylogeographic methods are clever. My key concern is about the multinomial regression. What were the other classes in the regression (0, 1 and?). Was time treated as a fixed or random effect? Was there evidence for homoscedasticity in the residuals as can be the case for GLMs with time series? Seems like more detail is needed to assess this analysis.

Our response: As discussed in the response to Reviewer 3 above, we added a more detailed explanation of our approach and code and data are provided to reproduce it. The model did not include random effects; time was the only covariate and was modelled as a fixed effect. The raw data modelled for each outbreak detected was which of the 5 lineages was “chosen”. Viruses were assumed to be independent (i.e., not competing with each other), which is reasonable given the low incidence of rabies in bats and that we are unaware of evidence for inter-lineage competition among rabies viruses. According to Li 2011 Transportation Research B and Abdulhafedh 2017 J Transportation Technology, the multinomial logistic regression does not assume homoscedasticity in residuals. Unfortunately, options for evaluating models are limited; however, we added a calculation of pseudo r-squared using the most conservative McFadden's method and p value for the effect of year to provide some indication of the fit of our model (P11, L267-269).

Initially, I was concerned that undetected virus circulation in the bats wouldn't be detected and this would be a significant limitation, but the arguments in the discussion satisfied my concerns. In fact, it may be good for the authors to stress this in the methods by adding a sentence or two.

Our response: We agree this is a good point to raise earlier. We now begin the methods section on putative invasions and extinctions with a statement that the aim is to evaluate evidence for extinctions given the possibility of undetected viral circulation (P9, L206-207):

“We evaluated evidence for viral invasions and extinctions given the possibility of undetected viral circulation using two distinct approaches.”

We also explicitly raise this point in the abstract (P2, L37-38):

“Statistical models suggested that lineage disappearances were more likely to be explained by viral extinctions than undetected viral circulation.”

Otherwise, most of my comments are relatively minor.

138: Why only one gene? For those unfamiliar with rabies, some explanation of why this small region (1500 bp) was targeted and some of the limitations with just using a small region of the rabies genome is needed.

Our response: We added a sentence to the methods section justifying our use of the nucleoprotein gene (P6, L136-140):

“The nucleoprotein is an informative gene for phyllogeographic analyses of rabies and is the most widely sequenced VBRV gene in Genbank (2440 records versus 596 in the glycoprotein and fewer in other genes; Accessed 31 July 2019 via <http://rabv.glue.cvr.ac.uk>), which maximized our ability to detect incursions into or out of Costa Rica [18,29].”

In the discussion, we added a sentence and reference on the limitations of using only one of the 5 rabies genes (P15, L362-364):

“Ideally, these datasets could adopt viral whole genome sequencing, which would be expected to increase the resolution of spatiotemporal inferences compared to the single-gene approach used here [48].”

162: Using separate substitution models for each codon position, to my knowledge, is not readily doable using the BEAUTI interface. How was this done?

Our response: This was accomplished by manually editing elements of the xml file. Briefly, this involves setting nucleotide substitution rates equal to each other and removing priors/operators on the parameters no longer needed in the simplified model. This can be achieved following a tutorial available on the BEAST website (https://beast.community/custom_substitution_models). We added a statement to the methods section (P8, L178-179):

“Customized nucleotide substitution models were implemented BEAST by manually editing the xml file generated by BEAUTI.”

170: A sup table with all of these BF and likelihood values is needed here.

Our response: Added as requested. This is now ESM File 1 and later ESM files were re-numbered.

173: In this case, why weren't too separate analyses performed?

Our response: We opted not to conduct two separate analyses due to the limited number of samples available (see P8, L188-193). This allowed us to leverage shared information across the two focal viral lineages, while allowing for independent tree topologies (see Suchard et al. 2003 Systematic Biology for a general introduction to hierarchical phylogenetic modelling and Streicker et al. 2013 PLoS Pathogens for an example using rabies). We believe that the benefits of this approach outweigh the potential benefit from considering additional clock models, especially since our analysis of the larger international dataset did not support the random local clock.

194: More detail is required here i.e. which particular statistic.

Our response: We used the mean branch velocity here since it would be most comparable to previous estimates. More detail has been provided in this sentence (P8, L198-200):

“Dispersal velocities (mean branch velocity) were calculated using 1000 randomly sampled, post burn-in trees from the posterior using the Seraphim package in R”

195-6: This detail is superfluous - could be dropped.

Our response: We removed this sentence. Interested readers can see how dates were modelled in the xml files.

196: Why 6 months?

Our response: Apologies if the wording was unclear, we modelled 6 months of uncertainty on both sides of the midpoint of the year creating one year of uncertainty in total. This is equivalent to setting precision to 1. The revised text has been simplified (P9, L200-201):

“We modelled one year of uncertainty on the date of samples for which only the year of collection was available (precision = 1).”

198: Why weren't ESS's checked? This seems like an obvious omission.

Our response: ESS's were checked and we have now added a statement on this to the methods section (P8, L201-203):

“We used Tracer v1.7.1 to verify parameter convergence, check that effective sample sizes on parameters exceeded 200, and select appropriate burn-ins.”

220: Just for the N gene correct?

Our response: *Thanks for catching this. We updated the text to be explicit that our rate estimate refers to the Nucleoprotein (P10, L236).*

Fig. 1c. Where is CR? Am I missing something?

Our response: *Viral dispersals involving Costa Rica are shown in panel B of the figure. We edited the caption to clarify that panel C only includes dispersals that were not already shown in panel B.*

Appendix C

Point-by-Point Responses to the Reviewers' Critiques

Reviewer(s)' Comments to Author:

Referee: 3

Comments to the Author(s)

Going through the revised manuscript, I realise that there were indeed specific methodological aspects that I misunderstood in study. While I think the text is clearer now, I also admit my lack of expertise to properly assess the novel analysis presented here (temporal analysis of putative national viral invasions and extinctions). However, I get the general idea and really appreciate reading this study joining phylodynamic (involving a relevant combination of discrete and continuous phylogeographic inferences) and modelling approaches. My only remaining general comment is that the link between the two (even if stated e.g. in line 222) should be even more explicit or put forward elsewhere (abstract?). Here are a series of other minor comments:

Our response: We revised the final sentence of the abstract to make the methodological advance of the study more explicit. It now reads: "...recurrent pathogen extinctions and re-invasions which can be readily detected in relatively small datasets by joining phylodynamic and modelling approaches."

- figure S1: I would actually use a continuous color scale to color the dots. Also, how do you refer/indicate the samples that were unavailable for sequencing? This is not obvious to me right now and I think that sequenced cases should be clearly highlighted on the map. And finally, another detail: what's the background map (I guess elevation...)? Please report its nature, a color scale and a source. This figure is actually important regarding the subsequent analyses performed on these outbreak records. This is of course only a matter of personal preference, but I would prefer such a modified version of Figure 1 rather than the current version of Figure 3.

Our response: We remade the figure using a continuous color scale, but given the large number of years shown, it was difficult to distinguish time by color alone. We therefore added a numerical key to each outbreak. We also used separate color scale to indicate the outbreaks that were sequenced. For Figure 3, we think the reviewer was suggesting adding information on the location of outbreaks that were not sequenced, which has now been added to panel A.

- figure 1: I really like it, but I would also really advise the authors to better highlight the sequences coming from Costa Rica (switching colors? coloring sequence names?). This should be directly obvious for the reader where these sequences are in the tree...

Our response: We have highlighted the sequences from Costa Rica by plotting tip labels bright red.

- figure 2: could add some kind of temporal information on it? This could potentially better illustrate spread directions.

Our response: We added points for the observed samples, which combined with the inferred geographic origin of each clade, illustrates the directions of spread for each virus.

- abstract: it remains not obvious to me what the authors exactly mean by "no lineage persisted was detected throughout the duration of surveillance".

Our response: We revised the text to read: "...and no lineage was detected across all years of surveillance."

Referee: 4

Comments to the Author(s)

The authors have done a good job of addressing my statistical concerns.

My only suggestion is to include a reference to Figure 4 line 271 to make it clear where the CIs are they are referring to.

Our response: The reference to the figure was added to the sentence as suggested.